# Extracting lung function measurements to enhance phenotyping of chronic obstructive pulmonary disease (COPD) in an electronic health record using automated tools

**Kathleen M. Akgün**[1,2]*, **Keith Sigel**[3], **Kei-Hoi Cheung**[1,4], **Farah Kidwai-Khan**[1,2], **Alex K. Bryant**[5], **Cynthia Brandt**[1,4], **Amy Justice**[1,2], **Kristina Crothers**[6,7]

1 Department of Medicine, VA Connecticut Healthcare System, West Haven, CT, United States of America,
2 Department of Medicine, Yale University School of Medicine, New Haven, CT, United States of America,
3 Division of General Internal Medicine, Icahn School of Medicine at Mount Sinai, New York, New York, United States of America, 4 Department of Emergency Medicine, Yale University School of Medicine, New Haven, Connecticut, United States of America, 5 Department of Radiation Oncology, University of Michigan, Ann Arbor, Michigan, United States of America, 6 Department of Medicine, VA Puget Sound Health Care System and University of Washington, Seattle, Washington, United States of America, 7 Department of Medicine, University of Washington School of Medicine, Seattle, WA, United States of America

* Kathleen.akgun@yale.edu

**Data Availability Statement:** All relevant data are within the paper and its Supporting Information files.

## Abstract

### Background

Chronic obstructive pulmonary disease (COPD) is associated with poor quality of life, hospitalization and mortality. COPD phenotype includes using pulmonary function tests to determine airflow obstruction from the forced expiratory volume in one second (FEV1):forced vital capacity. FEV1 is a commonly used value for severity but is difficult to identify in structured electronic health record (EHR) data.

### Data source and methods

Using the Microsoft SQL Server's full-text search feature and string functions supporting regular-expression-like operations, we developed an automated tool to extract FEV1 values from progress notes to improve ascertainment of FEV1 in EHR in the Veterans Aging Cohort Study (VACS).

### Results

The automated tool increased quantifiable FEV1 values from 12,425 to 16,274 (24% increase in numeric FEV1). Using chart review as the reference, positive predictive value of the tool was 99% (95% Confidence interval: 98.2–100.0%) for identifying quantifiable FEV1 values and a recall value of 100%, yielding an F-measure of 0.99. The tool correctly identified FEV1 measurements in 95% of cases.

**Funding:** Dr. Akgün was supported by the Yale Cancer Center pilot grant (P30CA016359-37S4). This work was also supported by the National Institute on Alcohol Abuse and Alcoholism [U24-AA020794, U01-AA020790, U01-AA020795, U01-AA020799, U10 AA013566-completed to ACJ]; National Institutes of Health, National Heart, Lung, and Blood Institute [K07CA180782 and R01CA210806 to KS; R01 R01CA173754 to KC] and in kind by the US Department of Veterans Affairs. Disclosure: The views expressed in this manuscript represent those of the authors and do not necessarily represent those of the Department of Veterans Affairs.

**Competing interests:** The authors have declared that no competing interests exist.

## Conclusion

A SQL-based full text search of clinical notes for quantifiable FEV1 is efficient and improves the number of values available in VA data. Future work will examine how these methods can improve phenotyping of patients with COPD in the VA.

## Introduction

Chronic obstructive pulmonary disease (COPD) is a lung disease commonly associated with smoking and is the third leading cause of death in the United States. A diagnosis of COPD requires documentation of airflow obstruction from the ratio of the forced expiratory volume in one second (FEV1) to the forced vital capacity from spirometry testing. The FEV1 is a measure of airflow limitation that reflects COPD severity and is routinely included in pulmonary function tests (PFTs). Low FEV1 is associated with decreased quality of life and increased risk for hospitalizations and mortality.[1, 2]

Unfortunately, PFTs are frequently performed on vendor equipment that may not be linked to the primary electronic health record (EHR) for patients who undergo these studies.[3] Unreliable linkage to the EHR makes recovery of PFT values such as FEV1 challenging for research. Advanced natural language processing (NLP) tools have been developed to ascertain asthma in clinical cohorts and to extract pre- and post-bronchodilator FEV1 for patients with asthma, but require evaluation of performance in patients with fixed airflow obstruction and on a national level.[3, 4] These barriers to accessing PFT results hinder our ability to assess pulmonary function on the scale necessary to develop standard phenotypes of COPD outside of dedicated observational studies and clinical trials. Phenotyping physiologic COPD based upon direct measurement of FEV1 is important for patients, clinicians and researchers to advance our understanding of the clinical burden of COPD. Physiologic phenotyping by extracting FEV1 data from the EHR could also enhance our understanding of other lung diseases.

The Veterans Health Information Systems and Technology Architecture (VistA) [5] is VA's EHR system and provides backend (command line) database support for clinical transactions. The VA Corporate Data Warehouse (CDW) [https://www.hsrd.research.va.gov/for_researchers/vinci/cdw.cfm], which provides a structured query language (SQL) interface, contains national patient data from VistA structured to allow diverse analysis and reporting. PFTs can be identified using Common Procedure Terminology (CPT) codes, entries into VistA, scanned documents into the EHR, transcriptions into progress notes or through combinations of these approaches. CPT codes only inform researchers that a PFT study was ordered and completed but do not include quantifiable measurements such as FEV1. While some VA PFT equipment directly uploads structured FEV1 measurements into VistA, software update requirements prevent most sites from contributing FEV1 measurements to VistA for subsequent data analysis. To fill this gap, automatic tools can be used to identify FEV1 in the EHR, making them useful for clinical research.[6] These tools can identify FEV1 values within structured or unstructured data but must be tailored to individual EHR systems.

Development of clinically meaningful phenotypes for patients with COPD depends on developing an accurate approach to extracting FEV1 measurements from varied sources and types of EHR data. There are a range of approaches to clinical text mining including the use of NLP and regular expressions.[7–9] In addition, relational databases like Microsoft (MS) SQLServer integrate full-text indexing and regular expression features into the SQL query engine. The advantage of this hybrid approach is that it queries and integrates both structured

data and unstructured text in the database without requiring it to be extracted to external software and tools. To this end, we developed a SQL-based method for extracting FEV1 values from a large collection of clinical documents to characterize COPD severity in a cohort of patients with COPD who had PFTs performed. We have adopted this approach because our document corpus is stored in a relational data warehouse (CDW that is implemented using the MS SQLServer) and the documents are well-defined enough so that we do not require a heavy-weight NLP approach that would require significant informatics overhead and has a steep learning curve.

## Materials and methods

### Participants

We used data from a nationally representative cohort of HIV-infected and uninfected Veterans enrolled in the Veterans Aging Cohort Study (VACS). We used VACS for this project because of several ongoing projects involving obstructive lung disease (that would benefit from PFT data) and also because all relevant EHR data was stored on local servers, facilitating the data extraction work. We included VACS participants from fiscal years 1996–2015. We then identified participants who had PFTs within the VA system using relevant CPT codes (94010, 94150, 94060, 94726–9; 93720–93722, 94240, 94260, 94350, 94360, 94370, 94720, 94725). VACS has been approved by the Institutional Review Boards of the VA Connecticut Healthcare System and Yale University School of Medicine, granted a waiver of informed consent, and deemed HIPAA compliant.

**Sources of FEV1.**

**CDW:** CDW stores raw clinical data for patients receiving care at any VA site. CDW stores clinical and administrative data in separate tables (for example, tables for demographics, pharmacy and medication fill data, laboratory data, inpatient utilization data). PFT tables are also available but require the PFT equipment and software maintain compatibility to be uploaded into the EHR. PFT values are typically obtained from private vendors that are not always compatible with the VA EHR following software upgrades and new PFT equipment. Approximately 80 VA clinical sites had contributed to PFT raw data until the mid-2000s; with software updates and changes to PFT equipment, the links between PFT equipment have been reduced since that time and remains variable. However, 68% of VACS participants were enrolled by 2005, allowing ample number of observations for this study.

**FEV1 Extraction using SQLServer:** To extract FEV1 measurements, we collected progress notes for the participants, which are stored as Text Integration Utilities (TIU) documents, in the CDW. We developed a two-step text processing approach based on MS SQLServer's full-text search capability and its string processing functions that offer regular-expression-like search features. We implemented this approach using MS SQLServer 2014. TIU documents for the VACS participants were first processed using the full-text (keyword) search supported by MS SQLServer. This provides a speedy first pass to query a large set of TIU documents to retrieve those documents containing the "FEV" keyword. Although we can do the same search using the SQL "LIKE" query operator, it would be very slow because it does not use indexes. Using full-text index (supported by MS SQLServer) can make queries of large amounts of textual data efficient. Other useful features supported by the full-text index include i) the ability to look for terms that are "NEAR" one another, ii) customization of stop words (noise words) that can be removed from the full-text index and iii) the use of thesaurus that allows the user to define synonyms as part of a full-text query. In the second step of our approach, the documents returned from the full-text search were further processed by using a number of string processing functions that support regular-expression-like features. We used these functions to process the documents to identify certain FEV1 string patterns (e.g., FEV-1, FEV1, FEV_1;

negation for "fever") and extract the first FEV1 numeric value within 20 characters of the FEV1 string pattern from the TIU documents. We used string functions like PATINDEX, CHARINDEX and ISNURMERIC as part of the queries to facilitate extraction of numeric values within a plausible range and in a format consistent with the reporting of FEV1 values (Fig 1) (S1 Data code).

## Assessing SQL tool performance

To evaluate this process, we generated a reference standard from a random subset of TIU documents. This subset was identified as part of the extraction tool development to address FEV1 values from TIU documents with more than one FEV1 entity in the text. A pulmonologist (KMA) reviewed 128/250 (51%) of records with multiple FEV1 entities and another 200 records in another random set of TIU documents. The pulmonologist determined the presence of an FEV1 measurement closest to the date of the progress note reviewed and recorded that value. We compared the FEV1 value identified by chart review to the value extracted by the SQL tool. We measured precision and recall of the tool relative to chart review.[10] We also compared results from the SQL tool for FEV1 value extracted with CDW data for FEV1 values to determine whether the SQL tool added substantially to the number of unique FEV1 measurements extracted.

## Results

PFTs were identified among 41,689 unique patients using CPT codes (n = 204,300 CPT codes). CDW data had a quantifiable FEV1 from 12,425 of these patients. The SQL tool identified an additional 3,849 patients with numeric values for FEV1 that could be used for phenotyping this population (Fig 2). Of note, these FEV1 values were otherwise unavailable for analyses. The SQL code is available through GitHub: https://github.com/keicheung/FEV1-extraction. The overall distribution of the FEV1 values extracted by the SQL tool was similar to the distribution of values from the reference standard (Fig 3A), suggesting acceptable performance for use of the tool in this population. Distribution of a sample of FEV1 values obtained from structured data in CDW compared with the values extracted using the tool are illustrated in Fig 3B.

In our reference standard subset (n = 199 documents, 180 unique patients), the SQL tool had a positive predictive value of 99% (95% Confidence interval: 98.2–100.0%) for identifying records containing quantifiable FEV1 values and a recall value of 100%, yielding an F-measure of 0.99. Extraction of these values yielded a correctly identified FEV1 measurement in 95% of cases. Disagreements between the tool and chart review resulted from the tool incorrectly identifying an "FEV" term where it expected a numeric value to be present but it was not; text including fever was the most common explanation for these disagreements. Cases where both reviewer and the tool identified quantifiable FEV1 values had excellent agreement (Spearman's coefficient = 0.99).

## Discussion

This demonstrates that FEV1 values can be extracted using SQL queries with excellent ascertainment and good accuracy from unstructured text-based documents generated by the VA EHR. The SQL tool increased the yield for the number of complete FEV1 values by 24% when added to structured data sources from CDW. Overall, an FEV1 value was found in 9% of PFTs by CPT codes, and 5,958/41,689 (14%) unique patients with at least one PFT.

Prevalence of COPD has largely depended on administrative data such as ICD-9 codes, self-report on national surveys or combinations of approaches using observational data, with

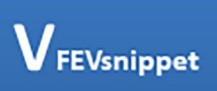

- **D_PFT** • Participants' documents with PFTs performed using CPT codes

- **D_FEVsnippet** • Documents (subset of D_PFT) containing 20 characters snippets with substring (FEV=, FEV 1, FEV -1, FEV- 1)

- **V_FEVsnippet** • Extract numeric values from each FEV snippet between 0.5-5.5L

**Fig 1. Algorithm implemented using SQL with fulltext search feature.** Algorithm supported by MS SQLServer 2014.

limited ability to measure severity.[11] Leveraging informatics tools for extracting FEV1 from the EHR can lead to a better-informed phenotyping of COPD and COPD severity in large patient populations, and to evaluate the effect of interventions. This could be used for large epidemiologic studies to characterize FEV1 or FEV1 trajectory over time and its association with clinical outcomes. This can also be used to identify potential COPD patients for non-pharmacologic and pharmacologic intervention aims at improving physical function and decreasing symptom burden for patients living with COPD.

Our SQL-based tool represents a light-weight but efficient approach to performing information retrieval/extraction on a large collection of clinical notes. This tool achieved desirable accuracy rates while maintaining the speed of data processing necessary for handling large

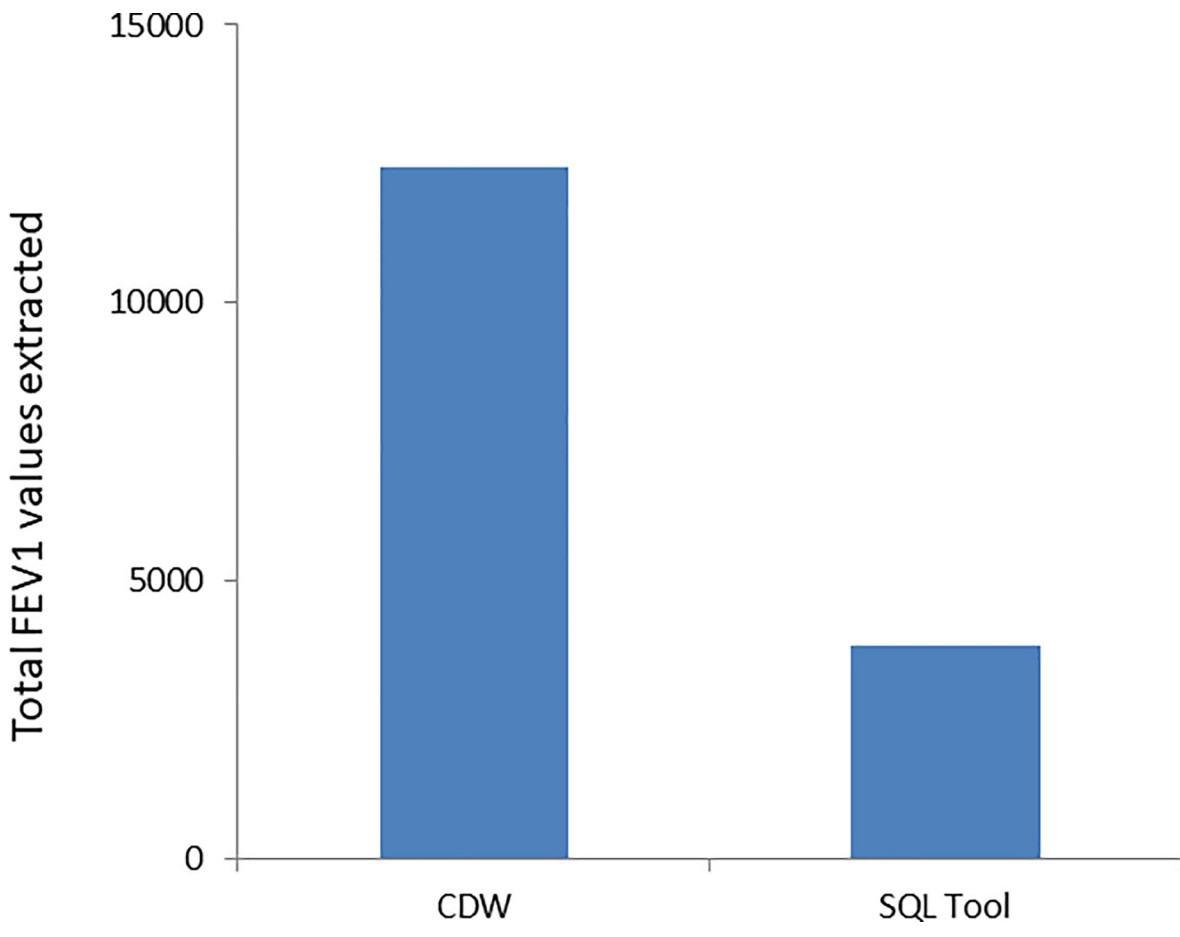

**Fig 2. Numeric FEV1 extraction yield including SQL tool.** The SQL tool increased FEV1 yield by 3849 (24%) compared with CDW alone.

data sets. Our tool's performance was not as sophisticated as previously developed NLP tools for extracting PFT results in the VA.[3] The approach developed by Sauer et al. to extract pre- and post-bronchodilator PFT results using semi-structured, unstructured and narrative text data in a sample of Veterans with asthma increased complete data by 25%. Another study using multiple clinical characteristics, including FEV1, had excellent agreement between a tool for asthma ascertainment and chart review.[4] Together, these tools demonstrate the range of informatics options available to improve characterization of lung disease from the EHR.

Despite its efficiency and accuracy, the SQL tool has some limitations. While the increase in FEV1 yield is similar between our study and NLP tools developed, it is important to highlight that we did not have a systematic approach for dealing with multiple FEV1 strings in one TIU document. The tool may have extracted an FEV1 value referring to a previous study rather than the current PFT study. However, we specifically evaluated the tool in documents with multiple FEV1 strings in them and found the performance to be good. In addition, the majority of PFT studies are completed using software external to the VA EHR operating system. PFT results are often scanned into the EHR. The SQL tool we developed and tested would not be able to extract FEV1 values from these external data sources. Our tool is also limited to the MS SQLServer implementation, as this implementation fits our purpose (i.e., the notes are stored in the CDW which is implemented using MS SQLServer). The string functions

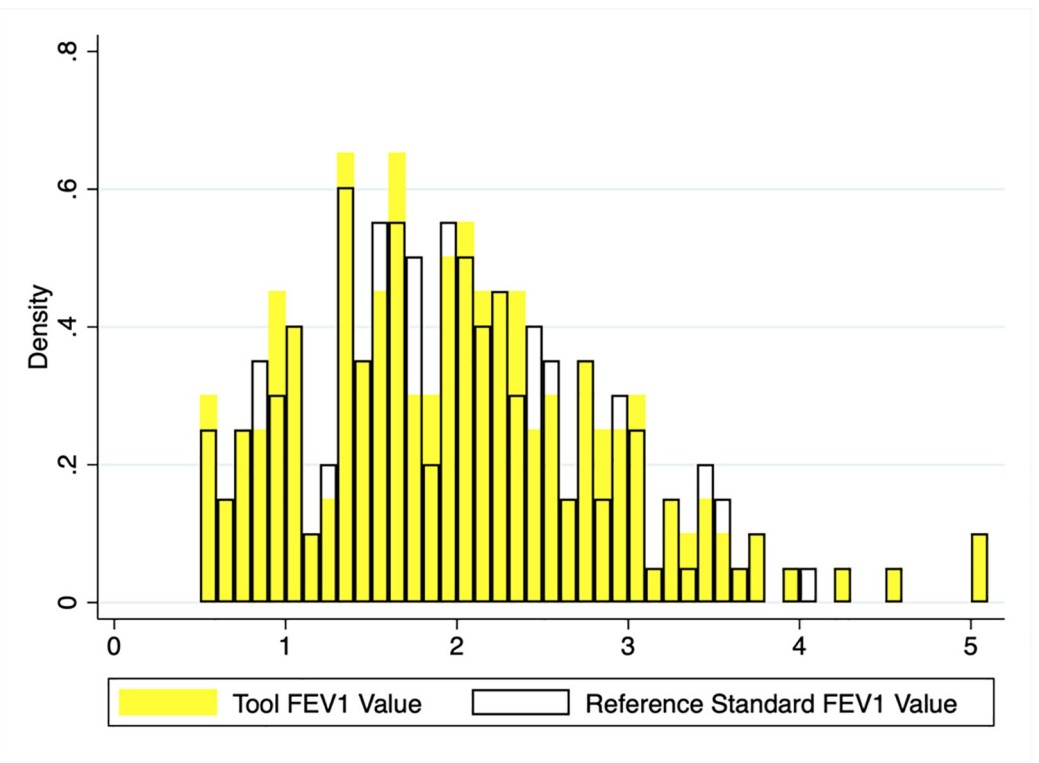

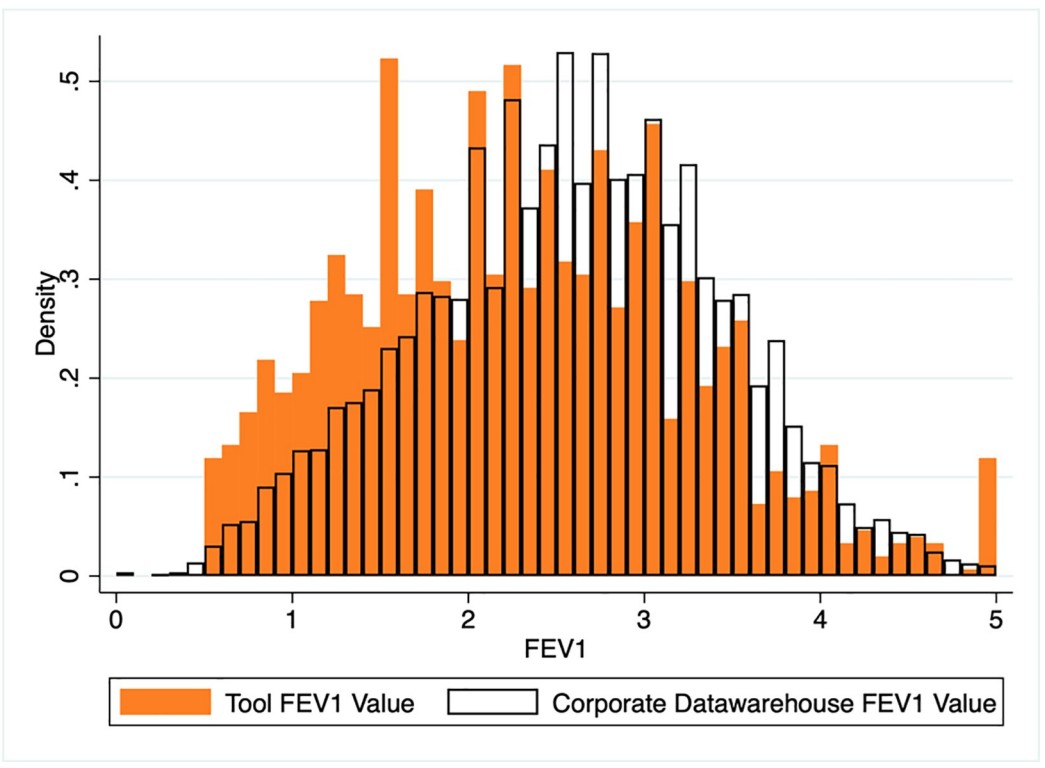

**Fig 3. a. Histogram of distribution of reference FEV1 values (clear shading) compared with the SQL extraction tool values (gold shading; n = 198 reference FEV1 values; n = 199 extraction tool values). Fig 3b. Histogram of the distribution of a sample of FEV1 values from structured Corporate Data Warehouse (CDW) tables and from extraction tool.** First FEV1 measurement for unique individual patients are shown (extraction tool, n = 1,510; CDW, n = 10,061).

supported by MS SQLServer 2014 do not offer the full-fledged regular expression capability. There remain patterns that our tool misinterpreted either as false positives or false negatives. Typically these patterns occurred rarely and it was therefore not practical to design specific rules to capture them on a case-by-case basis. In the future, we will explore a newer version MS SQLServer for new features that would help improve/enhance full-text searching and processing.

Our study has other limitations. First, this was only tested in one EHR from an already existing cohort of HIV-infected and uninfected patients. Second, we did not use extracted FEV1 values to estimate prevalence of COPD, and lack of identification of FEV1 does not necessarily imply absence of true COPD. Despite guidelines requiring FEV1 measurement for airway obstruction, spirometry may be performed in the minority of patients diagnosed with COPD, and airway obstruction is identified in only 56% of patients diagnosed COPD [12–16]. It is likely that there are systematic differences between patients who do and do not have FEV1 measurements completed, likely with the former also having more severe pulmonary disease. In addition, we did not include lung imaging, such as CT scan findings, another modality for measuring COPD severity.[17] Finally, while this does not include exhaled breathomics,[18] this extraction tool could be used to assist recruitment for COPD patients into subsequent studies, with potential candidates identified according to severity of FEV1.

Next steps will include characterization of data sources for patients who have CPT codes for PFTs but whose EHR does not include FEV1 measurements, namely, relying only on free text for FEV1 values, as well as developing and testing methods to extract data from scanned portable document format (PDF) or similar document types in the EHR. We will also explore how incorporating radiographic findings consistent with COPD affects performance of the FEV1 extraction tool alone. Future work is needed to develop the COPD phenotypes, apply these rules to other VA populations, and refine/test the SQL tool for FEV1 extraction and ascertainment outside of the VA EHR.

## Conclusions

SQL-based full text search of clinical notes can enhance the collection of quantifiable measurements of airflow limitations in VA data. Future studies are necessary to explore the use of similar techniques to extract data from non-VA EHR systems.

## Supporting information

**S1 Data code. SQL script used for extracting FEV1 values from text notes in VA clinical notes.**
(PDF)

## Acknowledgments

The views expressed in this manuscript represent those of the authors and do not necessarily represent those of the Department of Veterans Affairs.

**COI and Funding support:** Dr. Akgün was supported by the Yale Cancer Center pilot grant (P30CA016359-37S4). This work was also supported by the National Institute on Alcohol Abuse and Alcoholism [U24-AA020794, U01-AA020790, U01-AA020795, U01-AA020799, U10 AA013566-completed to ACJ]; National Institutes of Health, National Heart, Lung, and Blood Institute [K07CA180782 and R01CA210806 to KS; R01 R01CA173754 to KC] and in kind by the US Department of Veterans Affairs.

## Author Contributions

**Conceptualization:** Kathleen M. Akgün, Kristina Crothers.

**Data curation:** Farah Kidwai-Khan, Cynthia Brandt.

**Formal analysis:** Keith Sigel, Kei-Hoi Cheung, Cynthia Brandt.

**Funding acquisition:** Kathleen M. Akgün, Amy Justice.

**Methodology:** Kei-Hoi Cheung, Alex K. Bryant, Cynthia Brandt.

**Software:** Farah Kidwai-Khan, Cynthia Brandt.

**Supervision:** Cynthia Brandt, Amy Justice, Kristina Crothers.

**Writing – original draft:** Kathleen M. Akgün.

**Writing – review & editing:** Kathleen M. Akgün, Keith Sigel, Kei-Hoi Cheung, Farah Kidwai-Khan, Alex K. Bryant, Cynthia Brandt, Amy Justice, Kristina Crothers.

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
