## [Decision Letter · Decision Letter 0]

30 Sep 2019

PONE-D-19-20081

Extracting Lung Function Measurements to Enhance Phenotyping of Chronic Obstructive Pulmonary Disease (COPD) in an Electronic Health Record using Automated Tools

PLOS ONE

Dear Dr. Kathleen Akgun,

Thank you for submitting your manuscript to PLOS ONE. After careful consideration, we feel that it has merit but does not fully meet PLOS ONE’s publication criteria as it currently stands. Therefore, we invite you to submit a revised version of the manuscript that addresses the points raised during the review process.

Please resolve statistical analysis queries and make all data available. 

References about approaches for extracting FEV1 values from electronic health records, different or similar to that reported in the present study, should be reported and discussed.

Limitations of the study need to be implemented and conclusions more clearly described in the abstract for clincians.

We would appreciate receiving your revised manuscript by Nov 14 2019 11:59PM. To enhance the reproducibility of your results, we recommend that if applicable you deposit your laboratory protocols in protocols.io, where a protocol can be assigned its own identifier (DOI) such that it can be cited independently in the future. For instructions see: http://journals.plos.org/plosone/s/submission-guidelines#loc-laboratory-protocols

We look forward to receiving your revised manuscript.

Kind regards,

Manlio Milanese

Academic Editor

PLOS ONE

Journal Requirements:

2. Please provide the full name of the IRB that approved the VACS study.

Additional Editor Comments (if provided):

Please resolve statistical analysis queries and make all data available.

References about approaches for extracting FEV1 values from electronic health records, different or similar to that reported in the present study, should be reported and discussed.

Limitations of the study need to be implemented and conclusions more clearly described in the abstract for clincians.

Reviewers' comments:

Reviewer's Responses to Questions

**Comments to the Author**

1. Is the manuscript technically sound, and do the data support the conclusions?

Reviewer #1: Yes

Reviewer #2: Partly

Reviewer #3: Partly

2. Has the statistical analysis been performed appropriately and rigorously? 

Reviewer #1: Yes

Reviewer #2: No

Reviewer #3: No

3. Have the authors made all data underlying the findings in their manuscript fully available?

Reviewer #1: Yes

Reviewer #2: Yes

Reviewer #3: No

4. Is the manuscript presented in an intelligible fashion and written in standard English?

Reviewer #1: Yes

Reviewer #2: Yes

Reviewer #3: Yes

5. Review Comments to the Author

Reviewer #1: I will focus on methods and reporting. This is an excellent paper, with clear aims and implementation. The code has been made freely available through github, bravo!

I only have some minor comments which shouldn't take too long to deliver, to make the evaluation clearer.

1) Why don't you display the distributions of the available values and the extracted values in overlapping histograms.

2) a second histogram of the values extracted by the clinician against the ones extracted by the tool in the validation subsample (or of their differences).

3) you can report the mean bias (is there overall bias in the tool, say reporting values higher) and the mean square error as well, they can be more informative than the kappa.

Reviewer #2: The authors present an efficient method for extracting FEV1 from medical notes to improve phenotyping of Veteran patients with COPD. This work commendable because it leverages existing database tools to retrieve and extract concept-values pairs for FEV1 from clinical notes. The intent of the author is unclear in some sections but can be resolved with editing. I made several comments in the attached document but my primary concerns, which center around how the authors chose to analyze the data and communicate their finding are listed below.

1. Lack of justification for using agreement statistics vs. accuracy statistics with chart-review clearly defined as the reference standard. There is a lack of consistency and precision with the primary performance measure (agreement vs. accuracy). The reviewer believes that agreement statistics, such as Kappa, are justifiable if the goal is to determine whether or not the NLP tool is as reliable as two or more human reviewers. Performance measures, such as SE, SP, PPV, accuracy or F-measure, precision, and recall, should be considered since the tool is compared against one human reviewer described as the reference standard.

2. This approach will make it easier to discover additional problems with the analysis and compare to existing literature. For example, conditioning on the presence of FEV quantifiable values and reporting agreement when the values are known from chart-review is problematic, since it will bias the overall performance measures (e.g., SE and SP). Furthermore, it does not support a comprehensive evaluation that would include false positives, accuracy, etc. The distribution of extracted values should also be compared to chart-review findings

3. It is not clear how the tool addressed situations where references to historical FEV results are handled when presented with current tests. Is the goal to identify current FEV test results on specific visit dates? The overall goal is not clearly detailed.

4. The results that reference the CPT population vs. NLP extracted population are not clear.

5. There appears to be a misunderstanding about the CDW and VistA systems that needs to be corrected.

6. The discussion needs more detail regarding the possibility that many FEV1 (PFT) reports are scanned in as image files and not accessible in the TIU notes.

7. Finally, the findings from this study should be compared to other NLP studies of PFTs from within the VA if possible. It is not clear if performance is a trade-off for efficiency or if both the performance and efficiency of this method are superior to other NLP efforts to extract FEV from the medical notes.

Reviewer #3: General comments

This is an original study which reports on the implementation of an automated tool, based on the Microsoft SQL, for extracting FEV1 values from the data repository of the Veterans Aging Cohort Study implemented using the MS SQLServer.

This reviewer has some formal and methodological concerns.

Specific comments

Major

- References about approaches for extracting FEV1 values from electronic health records, different or similar to that reported in the present study, should be reported and discussed.

- More detailed methods to replicate the automated tool presented in the present study should be provided as online information.

- The spirometric reports usually include the parameter (i.e. the "string term") "FEV1" expressed as: 1. measured value, in liters; 2. calculated percent predicted value; 3. predicted value for the examined subject, based on sex, age and height. It should be clarified what "string term" for "FEV1" was selected/extracted by using the described automated tool. Indeed, the spirometric report of a single subject usually includes all these three "string term" of "FEV1" and the automated tool might have found three different quantifiable "FEV1" values for the same subject.

- FEV6 is a spirometric index used instead of "FEV1", in particular when performing office spirometry, and possibly stored as parameter of pulmonary function test. Were "FEV6" "FEV-6", "FEV_6" excluded as string patterns from the string processing procedure?

- The results of the validity of the automated tool performance should be better presented. Agreement/disagreement between the automated tool and chart review (performed by a single Pulmonologist) are based on only n=128 documents (51% of those available) (see the section "Assessing SQL tool performance").

- Results from the present study might support the usefulness of the presented automated tool for increasing the detection of quantifiable FEV1 values in those electronic health records which are implemented using MS SQLServer. This does not mean that the automated tool enhance the detection of COPD patients in electronic health records. The discussion section, in particular the paragraph dedicated to the limitations of the study and the conclusions, should be reviewed accordingly.

- The abstract lacks of conclusions.

Minor

- Figure 1. Provide a legend for the abbreviations and unit of measurement for the y axis.

6. PLOS authors have the option to publish the peer review history of their article (what does this mean?). If published, this will include your full peer review and any attached files.

Reviewer #1: No

Reviewer #2: Yes: Brian C. Sauer

Reviewer #3: No

---

## [Author Response · Author response to Decision Letter 0]

4 Dec 2019

Reviewer #1

Comments to the Author

I will focus on methods and reporting. This is an excellent paper, with clear aims and implementation. The code has been made freely available through github, bravo! 

I only have some minor comments which shouldn't take too long to deliver, to make the evaluation clearer.

1) Why don't you display the distributions of the available values and the extracted values in overlapping histograms.

Response: We have included Fig 3a according to this suggestion, and hope it will address the request of the reviewer that is perhaps hoping to make the comparison of the two methods more telegraphic.

2) a second histogram of the values extracted by the clinician against the ones extracted by the tool in the validation subsample (or of their differences).

Response: To complement the figure suggested in the previous comment, we have included a figure to address this suggestion, as well (Fig 3b).

3) you can report the mean bias (is there overall bias in the tool, say reporting values higher) and the mean square error as well, they can be more informative than the kappa.

Response: We are glad to hear the reviewer’s overall feedback on the manuscript. We have included new performance statistics for the tool based on comments from other reviewers. We do not report the mean square error, but do report the positive predictive value and its confidence interval (99% [98.2-100.0]). Importantly, we erroneously reported the performance of an earlier version of the SQL tool, and were able to include a larger reference standard (n=198 documents) than reported in the original submission. We hope these issues are clear throughout the manuscript and that the response is sufficient to address the reviewer’s comment.

Reviewer #2

Comments to the Author

The authors present an efficient method for extracting FEV1 from medical notes to improve phenotyping of Veteran patients with COPD. This work commendable because it leverages existing database tools to retrieve and extract concept-values pairs for FEV1 from clinical notes. The intent of the author is unclear in some sections but can be resolved with editing. I made several comments in the attached document but my primary concerns, which center around how the authors chose to analyze the data and communicate their finding are listed below.

Response: We have edited the manuscript to clarify the intent for this work throughout the revision. We intended to first develop and test the SQL tool to extract FEV1 values, with future work aimed at using this tool to then better characterize COPD severity in a large, national sample of patients. 

1. Lack of justification for using agreement statistics vs. accuracy statistics with chart-review clearly defined as the reference standard. There is a lack of consistency and precision with the primary performance measure (agreement vs. accuracy). The reviewer believes that agreement statistics, such as Kappa, are justifiable if the goal is to determine whether or not the NLP tool is as reliable as two or more human reviewers. Performance measures, such as SE, SP, PPV, accuracy or F-measure, precision, and recall, should be considered since the tool is compared against one human reviewer described as the reference standard.

Response: We included precision and recall performance of the tool, first referencing this plan in the methods, and then reporting in the results. Of note, the development and testing of this tool was iterative, and we had completed more chart reviews that originally reported for the initial submission, with tool performance significantly higher than reported with the initial submission. Thus, we report the PPV, recall and F-measure for the updated tool.

“In our reference standard subset (n=199 documents, 180 unique patients), the SQL tool had a positive predictive value of 99% (95% Confidence interval: 98.2-100.0%) for identifying records containing quantifiable FEV1 values and a recall value of 100%, yielding an F-measure of 0.99. Extraction of these values yielded a correctly identified FEV1 measurement in 95% of cases.”

2. This approach will make it easier to discover additional problems with the analysis and compare to existing literature. For example, conditioning on the presence of FEV quantifiable values and reporting agreement when the values are known from chart-review is problematic, since it will bias the overall performance measures (e.g., SE and SP). Furthermore, it does not support a comprehensive evaluation that would include false positives, accuracy, etc. The distribution of extracted values should also be compared to chart-review findings

Response: We hope this comment is at least partly answered with the inclusion of Fig 3a and 3b, with Fig 3b showing the distribution of the extracted values compared with the reference standard.

3. It is not clear how the tool addressed situations where references to historical FEV results are handled when presented with current tests. Is the goal to identify current FEV test results on specific visit dates? The overall goal is not clearly detailed.

Response: The reviewer raises important limitations of this approach to extracting spirometric values. These were concerns shared by the co-authors for this work. To begin to assess the risk multiple string values in one report might pose to the performance of the tool, we performed chart review for clinical notes with more than one FEV1 string value listed. We programmed the tool to extract the first FEV1 string in the progress note of interest. Thus, we are encouraged by the performance of the tool used based on the progress notes enriched with multiple FEV1 string terms. We have included text in the methods to clarify which string term was selected (highlighted section are the words added). 

(Methods section, Sources of FEV1, FEV1 Extraction using SQLServer, second to last sentence)

“We used these functions to process the documents to identify certain FEV1 string patterns (e.g., FEV-1, FEV1, FEV_1; negation for “fever”) and extract the first FEV1 numeric value within 20 characters of the FEV1 string pattern from the TIU documents.” 

(Discussion section, 4th paragraph)

“While the increase in FEV1 yield is similar between our study and tools, it is important to highlight that we did not have a systematic approach for dealing with multiple FEV1 strings in one TIU document. The tool may have extracted an FEV1 value referring to a previous study rather than the current PFT study. However, we specifically evaluated the tool in documents with multiple FEV1 strings in them and found the performance to be good.”

4. The results that reference the CPT population vs. NLP extracted population are not clear.

Response: We simplified the results section to make comparisons between the source population (i.e., those with CPT codes for PFTs) and the number of patients with FEV1 values from the source population in structured, CDW data compared with the yield using the SQL tool. We deleted sentences that included the number of documents that the SQL tool identified FEV1 entities, as well.

(Results section, 1st paragraph)

“PFTs were identified among 41,689 unique patients using CPT codes (n=204,300 CPT codes). CDW data had a quantifiable FEV1 from 12,425 patients of these patients. Using the SQL tool among patients identified as having PFTs using CPT codes (n=204,300 CPT codes, n=41,689 unique patients), an FEV1 entity was identified in 18,183 TIU documents among 5,958 unique patients. CDW data, in contrast, had a quantifiable FEV1 from 12,425 patients.”

5. There appears to be a misunderstanding about the CDW and VistA systems that needs to be corrected.

Response: We clarified the relationship between VistA and CDW throughout the manuscript, and included specific language regarding the two systems in the introduction.

(Introduction, 3rd paragraph)

“The Veterans Health Information Systems and Technology Architecture (VistA) [5] is VA’s EHR system and provides backend (command line) database support for clinical transactions. The VA Corporate Data Warehouse (CDW) [https://www.hsrd.research.va.gov/for_researchers/vinci/cdw.cfm], which provides an SQL interface, contains national patient data from VistA structured to allow diverse analysis and reporting.”

6. The discussion needs more detail regarding the possibility that many FEV1 (PFT) reports are scanned in as image files and not accessible in the TIU notes.

Response: We included additional statements in the discussion about this important limitation of this SQL tool. We also mention this as an area for future work.

(Discussion section, 4th paragraph)

“The majority of PFT studies are completed using software external to the VA EHR operating system. Results are often scanned into the EHR. These SQL tool we developed and tested would not be able to extract FEV1 values from these data sources.”

“as well as developing and testing methods to extract data from scanned portable document format (PDF) or similar document types s in the EHR.”

7. Finally, the findings from this study should be compared to other NLP studies of PFTs from within the VA if possible. It is not clear if performance is a trade-off for efficiency or if both the performance and efficiency of this method are superior to other NLP efforts to extract FEV from the medical notes.

Response: We include points of discussion of NLP tools that were developed for FEV1 extraction for patients with asthma and ascertainment (references discussed: Sauer BC et al, EGEMS 2016 and Wi CI et al, J Allergy Clin Immunol Pract 2018). We put our study into these contexts, but also to see these resources as opportunities to improve the quality of pulmonary-related clinical assessments from EHR data. Some of the added text is included.

(Discussion, 3rd paragraph)

“Our tool’s performance was not as sophisticated as previously developed natural language processing (NLP) tools for extracting PFT results in the VA.1 The approach developed by Sauer et al. to extract pre- and post-bronchodilator PFT results using semi-structured, unstructured and narrative text data in a sample of Veterans with asthma increased complete data by 25%. Another study using multiple clinical characteristics, including FEV1, had excellent agreement between a tool for asthma ascertainment and chart review.2 Together, these tools demonstrate the range of tools that can be used to improve characterization of lung disease from the EHR.”

Reviewer #3

Comments to the Author

This is an original study which reports on the implementation of an automated tool, based on the Microsoft SQL, for extracting FEV1 values from the data repository of the Veterans Aging Cohort Study implemented using the MS SQLServer.

This reviewer has some formal and methodological concerns.

Specific comments

Major

- References about approaches for extracting FEV1 values from electronic health records, different or similar to that reported in the present study, should be reported and discussed.

Response: We responded to a similar comment from another reviewer. Please see our response to Reviewer 2, Comment #7.

- More detailed methods to replicate the automated tool presented in the present study should be provided as online information.

Response: We certainly welcome the chance to have our approach replicated in other studies. We are under the impression making this available in github would be sufficient. If the reviewer and editor would prefer to have this as online supplemental material for the publication, we can prepare these resources.

- The spirometric reports usually include the parameter (i.e. the "string term") "FEV1" expressed as: 1. measured value, in liters; 2. calculated percent predicted value; 3. predicted value for the examined subject, based on sex, age and height. It should be clarified what "string term" for "FEV1" was selected/extracted by using the described automated tool. Indeed, the spirometric report of a single subject usually includes all these three "string term" of "FEV1" and the automated tool might have found three different quantifiable "FEV1" values for the same subject.

Response: We responded to a similar comment from another reviewer. Please see our response to Reviewer 2, Comment #3 to address this important potential limitations of the SQL tool used.

- FEV6 is a spirometric index used instead of "FEV1", in particular when performing office spirometry, and possibly stored as parameter of pulmonary function test. Were "FEV6" "FEV-6", "FEV_6" excluded as string patterns from the string processing procedure?

Response: We appreciate the comment. For this test case of extracting pulmonary function values from the electronic health record, we only searched for the FEV1. We have not found the FEV6 to consistently be used for clinical purposes. We suspected this string would be included in fewer notes and would limit our ability to extract this value from the clinical notes. However, based on what we have learned from FEV1 extraction, future work could focus on other PFT string values such as FEV6.

- The results of the validity of the automated tool performance should be better presented. Agreement/disagreement between the automated tool and chart review (performed by a single Pulmonologist) are based on only n=128 documents (51% of those available) (see the section "Assessing SQL tool performance").

Response: We respond to similar comments to this from other reviewers (please see Reviewer 2, Comment 1).

- Results from the present study might support the usefulness of the presented automated tool for increasing the detection of quantifiable FEV1 values in those electronic health records which are implemented using MS SQLServer. This does not mean that the automated tool enhance the detection of COPD patients in electronic health records. The discussion section, in particular the paragraph dedicated to the limitations of the study and the conclusions, should be reviewed accordingly.

Response: We attempt to respond to the concerns raised by this reviewer regarding the tool’s use, including that we have not used it to estimate prevalence of COPD. We made a number of changes to the discussion to clarify our work with the tool to date, and to specifically identify limitations on its use regarding estimates of COPD prevalence. We do not see areas in need of attention in the conclusions paragraph.

(Discussion 5th paragraph, with re-wording and deletions made to the paragraph).

“Second, we did not use extracted FEV1 values to estimate prevalence of COPD, and lack of identification of FEV1 does not necessarily imply absence of true COPD. Despite guidelines requiring FEV1 measurement for airway obstruction, spirometry may be performed in the minority of patients diagnosed with COPD, and airway obstruction is identified in only 56% of patients diagnosed COPD 3-7. This study does not address the potential for referral bias for patients who are having FEV1 measurements completed.”

- The abstract lacks of conclusions.

Response: We appreciate the reviewer pointing this out. We added two sentences to the abstract to conclude the results of this work at this stage.

“A SQL-based full text search of clinical notes for quantifiable FEV1 is efficient and improves the number of values available in VA data. Future work will examine how these methods can improve phenotyping of patients with COPD in the VA.”

---

## [Decision Letter · Decision Letter 1]

30 Dec 2019

Extracting Lung Function Measurements to Enhance Phenotyping of Chronic Obstructive Pulmonary Disease (COPD) in an Electronic Health Record using Automated Tools

PONE-D-19-20081R1

Dear Dr. Kathleen Akgun,

We are pleased to inform you that your manuscript has been judged scientifically suitable for publication and will be formally accepted for publication once it complies with all outstanding technical requirements.

With kind regards,

Manlio Milanese

Academic Editor

PLOS ONE

Additional Editor Comments (optional):

It is a pleasure to inform you that the article is now acceptable for publication on PlosOne.

Reviewers' comments:

Reviewer's Responses to Questions

**Comments to the Author**

1. If the authors have adequately addressed your comments raised in a previous round of review and you feel that this manuscript is now acceptable for publication, you may indicate that here to bypass the “Comments to the Author” section, enter your conflict of interest statement in the “Confidential to Editor” section, and submit your "Accept" recommendation.

Reviewer #2: All comments have been addressed

Reviewer #3: All comments have been addressed

2. Is the manuscript technically sound, and do the data support the conclusions?

Reviewer #2: Yes

Reviewer #3: Yes

3. Has the statistical analysis been performed appropriately and rigorously? 

Reviewer #2: Yes

Reviewer #3: Yes

4. Have the authors made all data underlying the findings in their manuscript fully available?

Reviewer #2: Yes

Reviewer #3: Yes

5. Is the manuscript presented in an intelligible fashion and written in standard English?

Reviewer #2: Yes

Reviewer #3: Yes

6. Review Comments to the Author

Reviewer #2: I believe the reviewers addressed the comments adequately. The one issue that continues to bother me is around the goal and framing of evaluation. It appears the goal of the SQL FEV extraction tool is to identify the "current" test result as of the date the note was written- meaning the most recent PFT on the date of the note vs. a co-referenced historical PFT result that may be typed or pasted into the document for historical reference with the "current" PFT. The authors target appears to be the current result up to the note date, but the tool appears to extract the FEV that appears first in the document. That is fine as a simplifying extraction assumption. The chart-reviewer/annotator was asked to abstract the current PFT results. In this case, accuracy statistics would say FEV was correctly identified but the mean values would not be the same. This is why including mean values or distribution of FEV values is important because you may correctly classify the document as having extractable FEV but extract the historical vs. the current. This can be revealed by the comparison of accuracy statistics with the distribution of FEV results. I would have expected the authors to explain why there was some inconsistencies in the extracted values to the chart-review values given the extremely high accuracy statistics. It is plausible that the differences in the distribution of values extracted from the tool and chart-review may be a result of grabbing the co-referenced "historical" FEV rather than the "current" FEV that the reviewer was asked to abstract.

It was also nice to see the distribution of FEV extracted from the tool overlay with PFT file data but I was initially thinking you would use the subset where both the FEV extraction tool and PFT file overlapped as additional validation or to identify records that may not have been reviewed but the tool picked them up - to determine if they were missed by chart-review due to the eligibility requirement of CPT coding that is problematic in this population. I the tool was run on the same eligibility criteria as chart-review then this is not an issue, but it was not previously clear what the note selection criteria was. The thought of a sub-analysis may be flawed anyways as there may not be a clear system ID linking a note to a PFT result and rules connecting the two would introduce additional error.

Overall this is excellent work and a more elegant solution than previous work in the VA. It is reassuring to learn that you are finding similar proportion of additional information from the notes that other studies found. It is fair to argue that other NLP approached in the VA m attempted different things and there may not be a direct comparison between your work and theirs. I do not believe the other approaches are more or less sophisticated, but what is in common is an attempt to extract information from notes that was not available in the PFT file since the PFT file is not complete. This approach appears more elegant and transportable than other approaches in the VA and is likely more valuable to the research community as a result.

Reviewer #3: (No Response)

7. PLOS authors have the option to publish the peer review history of their article (what does this mean?). If published, this will include your full peer review and any attached files.

Reviewer #2: Yes: Brian C. Sauer

Reviewer #3: Yes: Francesco Pistelli

---

## [Editor Report · Acceptance letter]

8 Jan 2020

PONE-D-19-20081R1 

Extracting Lung Function Measurements to Enhance Phenotyping of Chronic Obstructive Pulmonary Disease (COPD) in an Electronic Health Record using Automated Tools 

Dear Dr. Akgün:

I am pleased to inform you that your manuscript has been deemed suitable for publication in PLOS ONE. Congratulations! Your manuscript is now with our production department. 

With kind regards,

on behalf of

Dr. Manlio Milanese 

Academic Editor

PLOS ONE